# Metformin Treatment Reduces CRC Aggressiveness in a Glucose-Independent Manner: An *In Vitro* and *Ex Vivo* Study

**DOI:** 10.3390/cancers15143724

**Published:** 2023-07-22

**Authors:** Marie Boutaud, Clément Auger, Mireille Verdier, Niki Christou

**Affiliations:** 1UMR-INSERM 1308 CAPTuR, Faculté de Médecine, Institut OmegaHealth, Université de Limoges, 2 Rue du Dr Raymond Marcland, CEDEX, 87025 Limoges, France; marie.boutaud@unilim.fr (M.B.); mireille.verdier@unilim.fr (M.V.); 2Service de Chirurgie Digestive, Centre Hospitalier Universitaire de Limoges, 2 Av. Martin Luther King, CEDEX, 87000 Limoges, France

**Keywords:** colorectal cancer, metformin, glucose levels, *in vitro* study, patients’ cohort

## Abstract

**Simple Summary:**

In 2020, colorectal cancer (CRC) was ranked third among the most common cancers in the world, and second in terms of cancer-related mortality. Epithelial–mesenchymal transition (EMT) is mainly recognized by the loss of epithelial markers (such as E-cadherin) and cell movement activation, partially due to extracellular matrix remodeling by metalloproteinases, such as MMP2 and MMP9. It constitutes a critical step that promotes the spread of cancerous cells in oncogenesis, especially in epithelial cancers like CRC. Metformin is an anti-diabetic drug used in the treatment of type 2 diabetes; it targets mitochondrial metabolism and APMK. The EMT inhibitory effect from metformin used in the treatment of type 2 diabetic patients has been studied in large cohorts of patients with different cancer types; however, the mechanism of protection from metformin to colorectal cancer spread is still unknown, especially in the context of non-diabetes. Sortilin is a poor prognostic marker in CRC that may be related to signaling pathways that metformin could downregulate. This study focuses on the metformin-mediated effect on colorectal cancer aggressiveness according to different glucose conditions.

**Abstract:**

(1) Background: Metformin, an anti-diabetic drug, seems to protect against aggressive acquisition in colorectal cancers (CRCs). However, its mechanisms are still really unknown, raising questions about the possibility of its positive impact on non-diabetic patients with CRC. (2) Methods: An *in vitro* study based on human colon cancer cell lines and an *ex vivo* study with different colon cancer stages with proteomic and transcriptomic analyses were initiated. (3) Results: Metformin seems to protect from colon cancer invasive acquisition, irrespective of glucose concentration. (4) Conclusions: Metformin could be used as an adjuvant treatment to surgery for both diabetic and non-diabetic patients in order to prevent the acquisition of aggressiveness and, ultimately, recurrences.

## 1. Introduction

In 2020, colorectal cancer (CRC) was considered the third most frequent cancer and the second most deadly according to the WHO (World Health Organization). One of the major explanations for this high mortality rate relies on the stage at which it is detected, which favors therapeutic resistance and metastasis development [1]. In addition, the incidence rate has increased in the last few years following the enhanced prevalence of overweight, obesity, and metabolic diseases, such as type II diabetes mellitus, particularly in developed countries [2]. Such epidemiological data underline metabolic disorders as contributing factors for colorectal cancer development. At the same time, it appears that anti-diabetic treatments, such as metformin, a type II anti-diabetic drug called biguanide, showed beneficial effects in several cancers, including colorectal cancer [3]. Recently, in a retrospective cohort [4], we also reported that metformin use was associated with improved overall survival in diabetic patients with colorectal cancer. Indeed, patients who received metformin-based treatment had a 15.9% better chance of survival than type 2 diabetic patients who received other treatments after adjusting for confounding variables. These observations were also extended to other tumors, including gastric cancer [5], pancreatic cancer [6], medullary thyroid cancer [7], and endometrial carcinoma [8], where cell proliferation inhibition was reported. Metformin has also been shown to suppress tumor growth in animal models of ovarian cancer [9], melanoma [10], prostate cancer [11], and breast carcinoma [12].

Another anti-tumor effect of metformin relies on its ability to enhance chemotherapy efficiency, as reported by some previous studies. For example, Zhang et al. observed that metformin increases cisplatin-induced cell death through ROS generation in CRC cells [13]. Similar synergistic data were also reported between platinum drugs and metformin in ovarian cancer (for review [14]). Furthermore, it seems that its beneficial effects not only potentialize chemotherapy but also increase radiotherapy efficiency, particularly in CRC [15].

It is classically known that metformin reduces hepatic glycogenesis and, thus, increases glucose uptake into muscle cells in diabetic patients. Its anti-tumor effects could be directly linked to this effect on energetic status, especially on adenosine monophosphate kinase (AMPK) activation, through the inhibition of mitochondrial respiratory chain complex I, resulting in ATP synthesis reduction [16]. Indeed, this kinase is a key regulator of energy homeostasis due to its intervention in a great variety of metabolic pathways, such as the inhibition of the mTOR kinase, which activates autophagy and inhibits protein synthesis and cell growth. Furthermore, this activation of the AMPK pathway can prevent epithelial–mesenchymal transition (EMT), which is a mechanism used by cancer cells to migrate and establish metastasis [17]. The major hallmark of EMT is associated with the loss of expression of epithelial cell markers, such as E-cadherin. This transmembrane protein comes from multiple transcripts and various cleavages of a pre-protein that is imputable to several proteases, such as metalloproteinases MMP2 and MMP9 [18,19]. Among isoforms, the cleaved form of 30 kDa has been identified in previous studies as a poor prognostic marker in CRC [20]. *In vitro* studies have shown that metformin decreases EMT in certain cancers, such as gastric cancer [5] or colorectal cancer [21]. However, such *in vitro* studies, as well as patient cohort follow-ups, failed to highlight the potential of metformin, irrespective of the concentration of glucose, as clinical practice metformin is only used for diabetic patients. However, if the beneficial effect of metformin is attested, it could be of high interest to use it, even in non-diabetic patients.

Thus, the aim of this study was to evaluate the effect of metformin on colon cancer cell aggressiveness and in the patient cohort, irrespective of the concentration of glucose in cell lines or the diabetic status of patients.

## 2. Materials and Methods

### 2.1. Cell Cultures and Treatment

The human HCT-116 and SW-620 colon cancer cell lines were purchased from the ATCC (American Type Culture Collection, Manassas, VA, USA). HCT-116 and SW-620 are characterized as follows (Table 1) [22,23]:

Both cell lines were cultured in an RPMI medium (Gibco, Life Technologies, Cergy-Pontoise, France) supplemented with 10% FBS and 1% penicillin–streptomycin at 37 °C in a humidified atmosphere of 95% air plus 5% CO_2_. Two different glucose concentrations (1 g/L and 2 g/L) were applied, designated hereafter as low- and high-glucose conditions, respectively, and cells were acclimated to glucose levels for 2 weeks before further experiments were conducted. Cells were treated with different concentrations of metformin (Santa Cruz Tebu, Le Perray en Yvelines, France) for 24 to 72 h.

### 2.2. siRNA-Mediated Gene Silencing

AMP-activated protein kinase (AMPK) is an energy sensor that plays a crucial role in regulating cellular metabolism and maintaining energy homeostasis. Metformin has been shown to activate AMPK, which is believed to be responsible for many of the beneficial effects of metformin in treatments [24]. In order to evaluate AMPK’s contribution to metformin’s effects, we silenced its expression by siRNA. Cells were transfected with siRNAs using the DharmaFECT 1 transfection reagent (Horizon Discovery; T-2001-01, Cambridge, UK), according to the manufacturer’s instructions, with the recommended siRNA concentrations (25 nM). For AMPKα1 knockdown, a protein kinase-specific siRNA AMP-activated catalytic subunit alpha 1 (AMPKalpha1) was used (Horizon Discovery, L-005027-00-0010, Cambridge, UK). After two days of transfection, HCT-116 and SW-620 cells were treated or not with metformin at IC-50. Control siRNA (Santa Cruz Biotechnology, sc-37007, Dallas, TX, USA) was used at the same concentration as AMPK siRNA.

### 2.3. Cytotoxicity XTT Assay

The cytotoxic effect of metformin treatment was evaluated using an XTT assay (2,3-bis[2-methoxy-4-nitro-5-sulfophenyl]-2H-tetrazolium-5-carboxanilide; Cell Proliferation Kit II from Roche Diagnostics, Meylan, France). This test evaluates mitochondrial enzymatic activity by formazan production, which reflects cell functionality and (indirectly) cell viability. Moreover, $5\times 10^{3}$ cells/well were seeded into 96-well plates in the two different glucose conditions. The cells were treated with different concentrations of metformin (5, 10, 20, 30, 40, 60 mM) in triplicate or with RPMI in controls, and incubated for up to 24, 48, and 72 h. The XTT test was performed as recommended by the manufacturer. The absorbance at 490 nm (in reference to 690 nm) was determined in a Multiskan FC (Thermo Scientific, 11590685, Waltham, MA, USA) absorbance reader. The half-maximal inhibitory concentration (IC-50) of metformin on cell growth was chosen for further experiments.

### 2.4. Annexin V and Proliferation Assay

Apoptosis, cell survival, and proliferation were measured using the Incucyte S3 tool (Sartorius, Bohemia, NY, USA). Apoptotic cells were labeled with Incucyte^®^ Annexin V Dyes for Apoptosis (4642, Sartorius). The analysis was then performed using Incucyte^®^ Base Analysis Software (Incucyte S3 Software (v2018B)). We also quantified proliferation by counting the number of phase objects over time, using Incucyte’s cell-by-cell analysis software module. Moreover, $5\times 10^{3}$ cells/well were seeded into 96-well plates in the two different glucose conditions. The cells were treated with metformin at the 48 h IC50 concentration, and both apoptosis and proliferation were monitored over a 72-hour period.

### 2.5. Cell Migration

HCT116 cells, cultured in both glucose concentrations, were seeded at $6\times 10^{4}$ cells per well in a 96-well plate (Sartorius ImageLock, 4856), specific to the Incucyte device, and treated or not with metformin IC-50. Each condition was performed in technical triplicate. The wounds were made using the Incucyte^®^ 96-well WoundMaker tool. Monitoring of cell proliferation and migration was performed using Incucyte S3 (Sartorius). Nucleus labeling was achieved with Incucyte^®^ Rapid Red Dye for Live-Cell Nuclear Labeling (4717, Sartorius). The analysis was carried out with Incucyte^®^ Base Analysis Software. The proliferation was observed using Nuclight (Sartorius). The wound-healing density (%) was normalized by the red object’s total cell area (μm^2^).

### 2.6. Cell Invasion

HCT-116 cells were cultured in ULA plates at a concentration of 500 cells per well in a defined medium (DMEM-F12 medium containing 20 ng/mL EGF, 20 ng/mL FGF and 2(%) (*v*/*v*) B27) in order to allow the development of spheroids. Then, spheroids were included in 100 μL of Matrigel (100 μg/mL of RPMI medium with both glucose concentrations). The polymerized constructs were supplemented with 100 μL of culture medium treated or not with metformin at the 48 h IC50 concentration, per well. After at least 30 min of incubation, to allow the plate to reach optimal temperature, the ULA plate was placed in the Incucyte imaging system for real-time monitoring. A recommended 4× objective was used for imaging. Phase-contrast imaging was performed unless fluorescently labeled cells were used, and scans were conducted at intervals of 1 h.

### 2.7. Cohort of Patients

A total of 23 patients presenting colon cancer were included. Tumoral tissue samples from these patients were analyzed ranging from stage 1 to stage 3. Among them, 8 were non-diabetic and 8 were diabetic with metformin treatment. To obtain “control” tissue without any tumoral involvement, we analyzed samples from the adjacent non-tumoral tissue for each patient. The study was approved by the local ethical committee under the number 2022-007, and informed consent was obtained. The patients’ characteristics are summarized in Table 2.

Tissues were homogenized using TRIzol, and then proteins and RNA were extracted according to Direct-zolTM RNA Miniprep Kits (R2053, Zymo, Irvine, CA, USA) before being used for RT-qPCR and Western blot analysis (see below).

### 2.8. RNA Isolation and RT-qPCR Analysis

HCT-116 and SW-620 cells were treated with metformin IC-50 under low- and high-glucose culture conditions for 48 h. Total RNA was extracted using Direct-zolTM RNA Miniprep Kits (R2053, Zymo, Irvine, CA, USA), according to the manufacturer’s instructions. Moreover, 2 μg of RNA (from cell lines or tissues of patients) was subjected to cDNA synthesis using MultiScribe™ reverse transcriptase (4311235, Applied Biosystems, Fisher Scientific, Illkirch, France).

Real-time qPCR was performed in the QuantStudioTM instrument (Applied Biosystems) using the SensiFAST Probe Hi-ROX kit (BIO-82020, Meridian Bioscience, Cincinnati, OH, USA). Reactions were performed in triplicate from each biological replicate. Data were normalized to the ACTB transcript as reference genes and RNA expression levels were determined by the 2^−ΔΔCt^ method. The references of the primers are listed in Table 3.

### 2.9. Western Blotting

Total proteins, from control and metformin-treated cells grown under two different glucose conditions or from patients’ tissues, were extracted using RIPA buffer (50 mM Tris, 0.15 M NaCl, 1 mM EGTA, 1% NP40, 0.25% SDS, pH 7.4). The protein content of the cell lysate was quantified by the Bradford test (BioRad, Marnes-La-Coquette, France) and 30 μg was submitted to a 12% SDS-polyacrylamide gel and transferred to a polyvinylidene fluoride (PVDF; Amersham Pharmacia Biotech, Amersham, UK) membrane. After blocking non-specific sites with 5% BSA-TBS 0.1% tween-20 (1 h, RT), the membrane was incubated with the primary antibody (Table 4) in TBS-0.1% T-5% BSA overnight at 4 °C. After washing in TBS-T, the membrane was incubated with the appropriate secondary antibody conjugated to horseradish peroxidase (HRP) at 1:1000 dilution. Proteins were detected using Immobilon ECL Ultra Western HRP Substrate (Merck, Darmstadt, Germany) and revelation was achieved with G-Box system (Ozyme, Saint-Cyr-l’École, France). Alternatively, for normalization and expression quantification, we used a quantitative, fluorescent protein stain (TotalStain Q from Azure Biosystems).

### 2.10. Statistical Analysis

The IC-50 calculation was achieved thanks to the IC-50 calculation of GraphPad Prism, based on the following calculation:Fifty=(Top+Base)/2
$$Y=Bottom+\left(Top-Bottom\right)/\left(1+10^{\left((LogIC50-X)*HillSlope+log((Top-Bottom)/(Fifty-Low)-1)\right)}\right)$$

The statistical analysis of the confluence was performed using the ANOVA followed by multiple comparison tests of the AUC (area under the ROC curve) obtained after a 72 h culture. The statistical analysis of the *in vitro* RT-qPCR and Western blot was performed using the ANOVA followed by multiple comparison tests.

To analyze the wound closure of the cell migration, we used Incucyte^®^ Scratch Wound Analysis Software Module (cat. no. 9600-0012). The integrated analysis algorithm automatically masks each image to identify the position of the wounded (cell-free) and unwounded (cell-occupied) zones, to deliver robust measurements of wound width, wound confluence, and relative wound density (RWD) for the entire time course of the experiment.

The statistical analysis of the *ex vivo* RT-qPCR was realized using a z-score statistical test.

## 3. Results

### 3.1. In Vitro Study and Effects of Metformin

Since the main goal of the study was to investigate metformin’s effect on some CC aggressiveness markers in a glucose-independent manner, we performed an *in vitro* modelization of the diabetic cellular environment using two different types of glucose concentration media: low (1 g/L) and high (2 g/L). Before testing the metformin treatment effects, we checked the impacts of the culture conditions on cellular growth (Appendix A). Our results showed that glucose concentration did not impact cell growth during a 72 h culture.

#### 3.1.1. Cell Growth Inhibitory Effects of Metformin

Metformin treatment was previously reported to inhibit *in vitro* cell proliferation in various digestive tract cell models [25,26]. In our model, we calculated the minimal dose of metformin-induced toxicity of HCT-116 and SW-620 cells by metabolic testing, by incubating with concentrations from 0 to 60 mM in low- and high-glucose media. A cytotoxic test was performed 24, 48, and 72 h after treatment. As shown in Figure 1, the half-maximal inhibitory concentration (IC-50) of metformin in HCT-116 cell growth evolved from 36.78 mM in high-glucose media and 29.51 mM in low-glucose media after a 24 h treatment (Figure 1a) to 24.92 mM in high-glucose media and 21.48 mM in low-glucose media after a 48 h treatment (Figure 1c), and finally 6.59 mM in high-glucose media and 4.19 mM in low-glucose media after a 72 h treatment (Figure 1e). The same determinations were performed for SW620 cells: 34.16 mM in high-glucose media and 26.93 mM in low-glucose media after a 24 h treatment (Figure 1b); 20.86 mM in high-glucose media and 17.96 mM in low-glucose media after a 48 h treatment (Figure 1d); 12.32 mM in high-glucose media and 16.52 mM in low-glucose media after a 72 h treatment (Figure 1f). In both cell lines, the cellular morphology was not changed along the metformin treatment in low- and high-glucose conditions. These data show that metformin-inhibited cell growth is time- and dose-dependent for HCT-116 and SW-620.

For the following experiments, we worked with the 48 h-determined IC-50 concentrations of metformin, respective to glucose concentration and cell line.

#### 3.1.2. Metformin’s Effect on Apoptosis and Proliferation

To investigate the effects of metformin at 48 h with IC-50 on cell apoptosis and proliferation, an Annexin V assay was performed, along with cell-by-cell counting with the Incucyte tool. As shown in Figure 2a,b, HCT-116 and SW-620 cells failed to undergo apoptosis following metformin treatment under any glucose conditions. However, cell proliferation was strongly decreased when HCT-116 and SW-620 cells were treated with metformin irrespective of the glucose conditions (Figure 2c,d), indicating a cytostatic effect rather than a pro-apoptotic effect.

#### 3.1.3. Metformin’s Effect on E-Cadherin, MMP2, MMP9, Sortilin, and LC3-II Expressions with Different Glucose Concentrations

As previously reported [21], metformin was shown to reduce EMT at the same time as activating the AMPK pathway [27]. Thus, we focused on the E-cadherin expression in order to analyze the EMT process and on LC3-II conversion, an autophagic process marker, known to be amplified downstairs of AMPK activation. To complete the EMT analysis, we investigated the expressions of metalloprotease MM2 and MM9, known to be implicated in E-cadherin maturation and extracellular matrix remodeling [19,28]. In addition, we checked the expression of sortilin, whose association with aggressiveness and bad prognosis has been described in colorectal cancers (CRC) [29].

As shown in Figure 3a, the transcriptional analysis of CDH1 and SORT1 expressions exhibited different variations according to the cell line stage (early vs. advanced) and glucose concentration. Indeed, for HCT-116 cells, CDH1 was upregulated 3.097-fold in high-glucose medium and 1.863-fold in low-glucose after 48 h of treatment with IC-50 metformin. At the same time, SORT1 expression was not significantly affected either in high-glucose or low-glucose conditions. On the contrary, in the SW-620 cell line, CDH1 expression was not altered in the glucose-rich or -deficient medium after 48 h of metformin treatment whereas SORT1 expression was significantly increased (respectively, by 1.599 and 2.19 in high- and low-glucose conditions). These data showed a modified gene expression profile following metformin treatment.

Then, the protein expressions of the desired genes CDH1, MMP2, MMP9, and SORT1, as well as the LC3-B conversion, were evaluated (Figure 3a), and quantification was performed in reference to the overall protein expression evaluated by TotalStain Q (Appendix A). Furthermore, the Western blot analysis allowed us to distinguish between the total (120 kDa) and truncated (30 kDa) forms of E-cadherin. This last truncated form was largely expressed in non-treated early-stage HCT-116 cells and significantly decreased with the metformin treatment. In parallel, the same was observed with both MMP2 and MMP9 expressions, irrespective of the glucose concentrations, strengthening the link between the MMP and E-cadherin cleavage. The conversion of LC3-II increased in response to the treatment, which is consistent with the activation of APMK and, subsequently, autophagy, on metformin. In this cell line, the only protein that seemed to not be affected by metformin was sortilin. The protein change patterns were similar in both types of glucose culture media, underlying the effects of metformin independently of glucose concentration. Altogether, such data underlined the aggressiveness decrease along with metformin treatment in the early colon cancer cell line, irrespective of the glucose concentration. On the contrary, it seemed that the metformin treatment failed to modify the tested protein expression in the SW-620 cell line, irrespective of the glucose concentration, which reflects an advanced tumor stage.

#### 3.1.4. Analysis of the AMPK Role in the Metformin Inhibition of EMT According to Two Glucose Concentrations

The AMP-activated protein kinase (AMPK) pathway plays a vital role in cellular metabolism and energy homeostasis [16,30]. Interestingly, emerging research has identified a connection between AMPK, epithelial–mesenchymal transition (EMT), and the widely used diabetes drug, metformin [17]. Upon metformin treatment, AMPK activity is enhanced and, in turn, suppresses EMT, impeding tumor progression. In order to investigate the role of AMPK in metformin-mediated effects in both models, we underwent silencing of this kinase by siRNA. Therefore, we observed the effect of AMPK silencing on the protein expression of the desired genes CDH1, MMP2, MMP9, and SORT1, as well as LC3 conversion with or without metformin treatment (Figure 4). Quantification was performed with reference to overall protein expression assessed by TotalStain Q (Appendix A). For HCT-116 cells treated under both glucose conditions (Figure 4a,b), the AMPK siRNA inhibited most of the effects of metformin previously observed (Figure 3). Indeed, MMP2 was significantly less decreased in metformin-treated AMPK siRNA cells than in cells treated with metformin without siRNA or with a control siRNA (Appendix A). A similar pattern was observed for MMP9 even if it failed to reach a significant level. The metformin-mediated decrease in E-cadherin cleavage, as well as the increase in its total form, were also significantly reversed in AMPK siRNA-treated cells compared with cells treated with metformin without siRNA or with the control siRNA shown in Appendix A. Finally, the HCT116 cells, silenced for AMPK expression and treated with metformin, adopted behavior similar to control untreated cells. Since metformin had no effect on protein levels in SW-620 cells (Figure 3), AMPK siRNA failed to modify the metformin issue in SW-620 cells under both glucose conditions (Figure 4c,d).

#### 3.1.5. Metformin Inhibits Cell Migration and Invasion Regardless of Glucose Concentration

Loss of adhesion between cells allows cell migration, which is the first step of EMT and metastasis. The effects of metformin on cell migration were assessed by the wound-healing assay in the HCT-116 cell line, which is representative of the early stage and which is eager to spread.

Cells were grown until 90% confluence, scraped to create a wound, and closure was followed with Incucyte in treated and untreated (control) cultures for three days. As shown in (Figure 5), the control culture repaired the scar rapidly (within 48 h), whereas, in the metformin-treated samples, the healing process and closure were slowed. A comparison of the healing results with two concentrations of glucose indicated that the inhibition of cell migration by metformin was not affected by glucose levels. Wound closure was normalized by the number of nuclei labeled by Nuclight (Sartorius). This allowed us to conclude that the closure was due to migration and not proliferation. Furthermore, we observed that this wound closure was metformin-dose-dependent in addition to being time-dependent (Appendix A).

Another characteristic of an EMT feature consists of tumor invasion. In order to observe the effects of metformin on this process, we created spheroids from HCT-116 cells and embedded them in Matrigel. The invasion test was carried out for three days after inclusion under treated and untreated conditions and the spheroid size was followed with an Incucyte device. We observed a decrease in spheroid invasion, irrespective of the glucose conditions (Figure 6a). The sizes of treated spheroids failed to increase after 37 h, while those of untreated spheroids continued to grow (Figure 6b). In addition, the invasion was significantly reduced by around 2000 μm^2^ in both glucose conditions (Figure 6c). A comparison of the healing results between the two concentrations of glucose indicated that the inhibition of cell invasion by metformin was not affected by glucose levels.

### 3.2. Ex Vivo Study: Effects of Metformin Evaluated on Tissue Samples of Diabetic Patients at Different Colon Cancer Stages

Since our initial goal was to test metformin’s beneficial effects on colon cancer cells, independent of glucose metabolism status, we performed a preliminary study on samples from patients exhibiting different colon cancer stages, either diabetics treated with metformin or non-diabetic without anti-diabetic treatment. For the control, non-tumoral peripheral tissues to the tumor were analyzed.

#### 3.2.1. Metformin’s Effect on Gene Expressions of E-Cadherin and Sortilin

The changes in the expressions of tested markers were quantified at the mRNA level by real-time PCR. As shown in Figure 7a, CDH1 expression in patients with colon cancer treated by metformin appeared to revert to the expressions of non-tumoral tissues in stages 1 and 2. However, it seems to be the opposite in stage 3. Regarding SORT1, we failed to observe a return to an expression similar to that of “healthy tissue”, except to a minor extent, in stage 1; see Figure 7b. However, only a trend remains, as the z-score is not significant, likely due to the low sample size.

#### 3.2.2. Metformin’s Effect on Protein Expressions of E-Cadherin, Sortilin, and LC3-II

In addition to the transcriptional analysis, the expressions of the two forms of E-cad and sortilin were evaluated by Western blot as well, as the conversion of LC3-II. The protein ratio of E-cadherin (120 kDa) to E-cadherin (30 kDa) in the tissues of metformin-treated CC (colon cancer) patients, appeared to revert to an expression similar to healthy tissues in stages 1 and 2 but not in stage 3 (Figure 8 and Figure 9a). Such results corroborate data from transcriptional studies (Figure 7). The expression of sortilin seemed to decrease in metformin-treated CC patients in stages 1 and 2 (conversely to healthy tissues) but not in stage 3 (contrary to healthy tissues), in which it increased. As expected, the LC3-II expression increased in metformin-treated CC patients, especially in stages 1 and 2, which is coherent with APMK activation in metformin; Figure 9b,c.

## 4. Discussion

Colorectal cancer (CRC) is a deadly and highly prevalent cancer worldwide; it is one of the most frequent cancers in both genders [31]. The poor prognosis relies on late detection, which often occurs at the advanced/metastatic stage. In developed countries, another hindering factor links CRC with metabolic syndrome, especially type 2 diabetes mellitus [32,33]. Indeed, several epidemiological studies have underlined that diabetes-associated features, such as hyperglycemia, inflammation, and oxidative stress are risk factors for CRC development (for review [2]).

The classical therapeutic care of type 2 diabetes relies on metformin, a biguanide hypoglycemic class agent; epidemiological studies conducted by our team and others [3,4,34] revealed its beneficial effects on the oncological outcomes of CRC in diabetic patients, including the reduction of metastasis and overall survival. Nevertheless, these studies, by way of definition, are focused on the use of metformin in the context of hyperglycemia. In addition, an *in vitro* analysis of the mechanistic effects of the drug was also performed in hyperglycemia media (i.e., 2 g/L for RPMI medium), which represents the conventional glucose concentration for cell culture. Very few studies have targeted the influence of glucose growth conditions on metformin’s performance and opposite effects have been reported, possibly depending on the cancer cell type [5,26,35]. In addition, it is worth noting that the literature encompasses different studies, both clinical and experimental, which do not differentiate colon and rectal cancers, which are known to be anatomically and mechanistically completely distinct [36].

This prompted us to undertake a study on the effects of metformin, according to two different glucose concentrations, 1 g/L or 2 g/L, in two different colon cancer (CC) stage cell lines, i.e., early stage (HCT-116) and locally advanced stage (SW-620). Once cells were adapted to glucose-level culture conditions (2 weeks), this parameter failed to affect the cellular growth in both cell lines, conversely to the results from [26]. These authors showed an incidence of glucose concentration on cell proliferation. However, they used a higher glucose concentration than we did, i.e., 5 g/L, and cells that were metabolically different for glucose use in relation to the GLUT1 expression. In our study, the low concentration of glucose in the medium affected the cell sensitivity to metformin, since the calculated IC-50 was always lower in 1 g/L of cultured cells compared to 2 g/L of cultured cells at the three tested times, i.e., 24 h, 48 h, and 72 h (respectively 21.48 and 24.92 mM for HCT116 and 17.96 and 20.86 mM in SW620). Similar results were reported for ovarian cancer cells, where the enhancement of metformin cytotoxicity was exhibited in a low-glucose medium [35]. In addition, when testing apoptosis engagement in response to metformin, we did not observe any enhancement of this death pathway, indicating that, in our conditions, the drug is more prone to being cytostatic rather than pro-apoptotic. Such results are consistent with the wide range of metformin effects, from reduced proliferation to apoptotic or autophagic death [37,38].

Tumoral aggressiveness could be addressed by the investigation of EMT and associated parameters. Among them, the E-cadherin total and truncated forms, metalloproteinases involved in its cleavage, such as MMP2 and 9, and cellular migration and invasion capacities. E-cadherin belongs to the large family of adherence proteins and represents the prototypical cadherin of epithelial tissues [39]. When tumoral cells gain aggressiveness and engage in the metastasis process, an early event involves the loss of E-cadherin expression, which represents a hallmark of EMT [40]. In addition to the expression of the total form (120 kDa), some cleavage can also occur, releasing diverse truncated secreted or intracellular forms [41]. Such cleavages appeared to perturb the physiological role of E-cadherin to maintain cellular adhesion, by reducing the total form expression, thus leading to EMT. Indeed, in a previous study, we showed that cleaved forms of E-Cadherin were also associated with CRC progression [20]. Several enzymes are known to be responsible for this cleavage, including secretases, cathepsins, or matrix metalloproteases [42]. The latter is also involved in matrix remodeling, which contributes to the promotion of EMT and metastasis and, consequently, increases aggressiveness [43]. We showed an enhancement of CDH1 transcription and variations of post-translational modifications, with a great decrease of the p30 truncated form, upon treatment, especially in early stages, both *in vitro* (irrespective of the glucose concentration) and on the samples of patients (Figure 3, Figure 7 and Figure 9). This cleavage reduction was linked with the increased expression of the total form, which is coherent with the elevation of the adherent behavior of metformin-treated cells. The observed decreased expressions of MMP2 and MMP9 supported the reduced cleavage. Furthermore, it has been previously reported that AMPK activation, subsequent to metformin treatment, could inhibit the expressions of MMP-2 and MMP-9 [44]. The analysis of cellular migratory and invasive capacities underlined these results since HCT-116 cells (early stage) were drastically reduced when metformin was applied (Figure 5 and Figure 6). The reduction in aggressiveness upon metformin was also noticed to a lesser extent by the discrete reduction of sortilin expression, mostly in the early stages. This protein is known as a regulator of protein trafficking and sorting and as a receptor of neurotensin and neurotrophins [45]. Its role in cancer was reported to be controversial, depending on the cellular type and partners involved. In our previous experience, we observed that sortilin expression was upregulated in advanced stages, which made it a bad prognosis marker [29]. Here, metformin treatment led to a weak decrease in sortilin expression, contributing to the reduction of aggressiveness. Once again, Once again, a key finding was the independence of glucose concentration in the effects of metformin.

Linked to metformin-dependent APMK activation, we also observed LC3-II conversion, a classical marker of autophagy, irrespective of the glucose concentration. Indeed, this kinase, which acts as a sensor of the metabolic status of the cell, is known to regulate several cellular pathways, such as autophagy [16,30]. In accordance with our findings, ref. [46] recently reported on the enhancement of several autophagy-associated proteins, such as Beclin-1 and LC3-II, concurrent with a reduction in p62, when oral squamous cell carcinoma was treated with metformin. In order to deeply explore the implication of AMPK in the metformin effects, we evaluated the expressions of proteins of interest (E-cadherin, sortilin, LC3, MMP2, and MMP9) after silencing APMK by siRNA. As expected, the reduction of APMK expression in HCT-116 cells reversed the beneficial effects of metformin, i.e., reinforced the expression of metalloproteinases and, consequently, reestablished the cleaved form of E-cadherin (30 kDa) to the detriment of the total form. The link between AMPK and metalloproteinase activity was previously reported in physiological and pathological conditions [47,48]. Such a relation could explain our results and supports that, in our model, metformin deeply enhanced APMK activity, which reduced CRC aggressiveness. In the SW-620 cell line, since metformin failed to affect the cell behavior, the silencing of AMPK was also inefficient.

Parallelly, we also conducted a preliminary *ex vivo* study with tissues from patients with CC. Although our data need to be confirmed on a more extensive patient cohort, it seemed that only in the early stages of CC (Stages 1 and 2), E cadherin (both protein and transcripts levels were increasing) and sortilin expression (both protein and transcripts levels were decreasing) exhibited similar values in diabetic CC patients treated with metformin compared to healthy ones. Even very few studies dealing with metformin’s effects, according to CRC stages, our data are promising, as they are in line with previously published studies showing the benefits of metformin treatments, mainly in the early stages [49], or as a preventive agent for CRC development [50]. With these last clinical results, and taking into account the *in vitro* study, metformin may be used as a prophylactic drug in preventing colon carcinogenesis.

Overall, our study emphasized the importance of the stage at which the treatment was administered. Both *in vitro* and *ex vivo* data underlined that metformin was more prone to reducing aggressiveness criteria in the early stages. This may be because metformin reduces stemness [25] concomitantly with EMT [17,21], which is preferentially initiated by early-stage tumor cells.

## 5. Conclusions

In summary, our study showed the beneficial effects of metformin in reducing colon cancer progression. This noteworthy result relied on the independence of glucose concentration and emphasized the effects mainly in the early stages. Although such observations need to be confirmed, this bodes well for adjuvant treatment with metformin in the early stages of cancer in both diabetic and non-diabetic patients. 

## Figures and Tables

**Figure 1 cancers-15-03724-f001:**
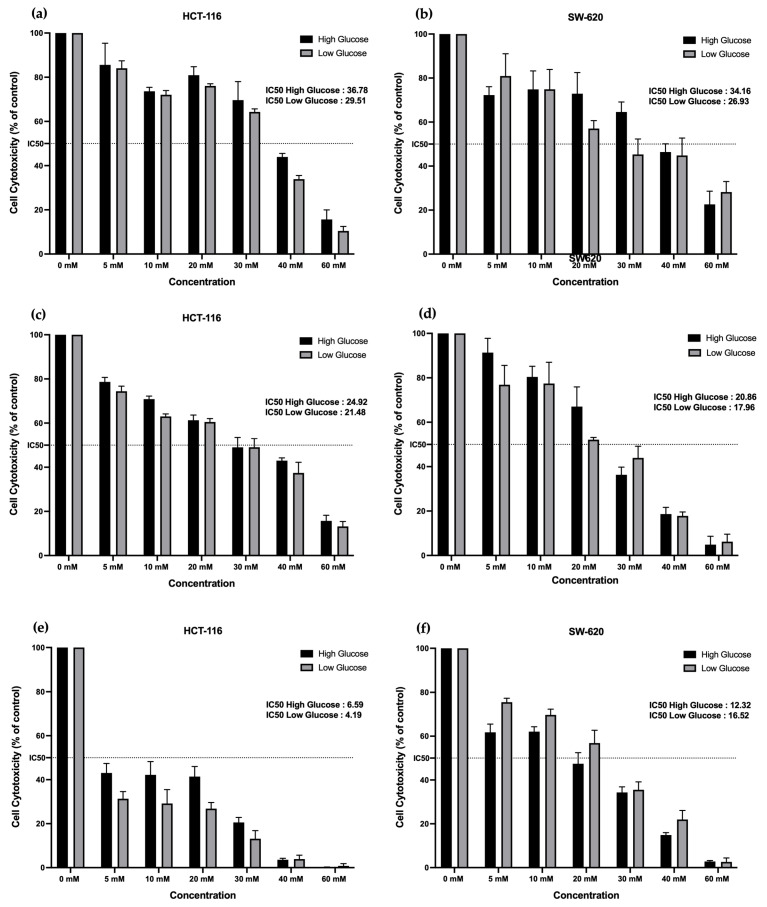
Cell toxicity of metformin in two glucose concentrations in HCT-116 and SW-620 cells. Concentrations of metformin varied from 0 to 60 mM in low- and high-glucose media. The inhibitory effects on cells were measured with XTT assay at 24, 48, and 72 h. Each bar graph shows the mean and SD of three independent experiments, performed in triplicate. HCT-116: (**a**) 24 h, (**c**) 48 h, (**e**) 72 h. SW-620: (**b**) 24 h, (**d**) 48 h, (**f**) 72 h.

**Figure 2 cancers-15-03724-f002:**
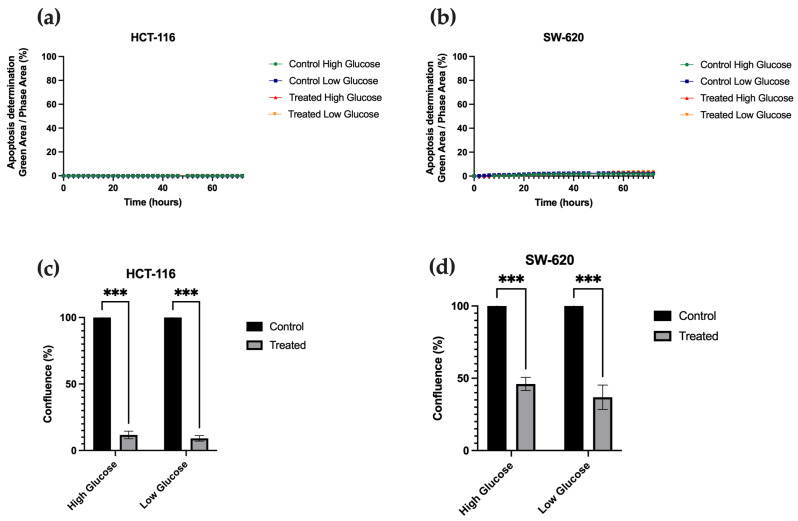
Metformin’s effect on apoptosis and proliferation in HCT-116 and SW-620. Detection of percentage apoptosis every hour in HCT-116 for 72 h in HCT-116 (**a**) and SW-620 (**b**) using the Incucyte tool. Areas under the ROC curve (AUC) were estimated to assess the ability of the cell confluence every hour for 72 h in HCT-116 (**c**) and SW-620 (**d**), using cell-by-cell analysis (Sartorius). *** *p* < 0.001.

**Figure 3 cancers-15-03724-f003:**
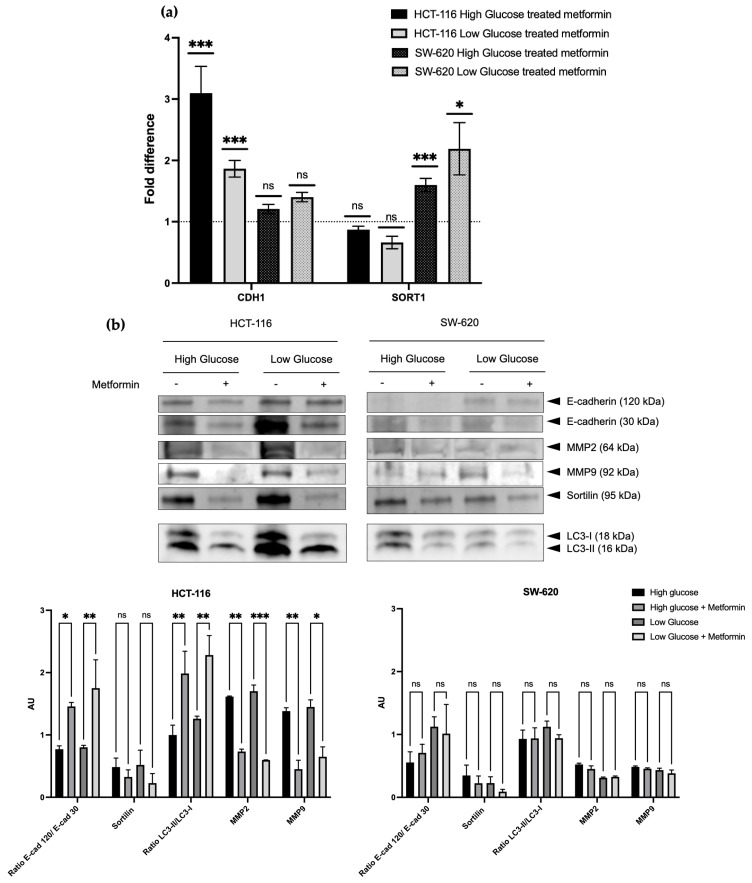
Metformin’s effect on E-cadherin, MMPs, and sortilin expressions in two glucose levels in HCT-116 and SW-620. (**a**) Transcriptomic analysis: indicated genes are quantified by real-time qPCR, after treatment with IC 50 metformin for 48 h in two glucose concentrations, in reference to ACTB for normalization. Each column shows the mean and SD of three independent experiments, performed in biological triplicate. The dotted line represents the normalization of untreated cells. (**b**) Proteomic analysis: The expressions of the indicated proteins were assessed via Western blot analysis on cells treated with IC50 metformin for 48 h in two different glucose concentrations. The TotalStain Q, (Appendix A) allowed us to evaluate the equal loading of protein. Performed in biological triplicate. * *p* < 0.05; ** *p* < 0.01; *** *p* < 0.001. ns: no significance. The uncropped blots are shown in Appendix A.

**Figure 4 cancers-15-03724-f004:**
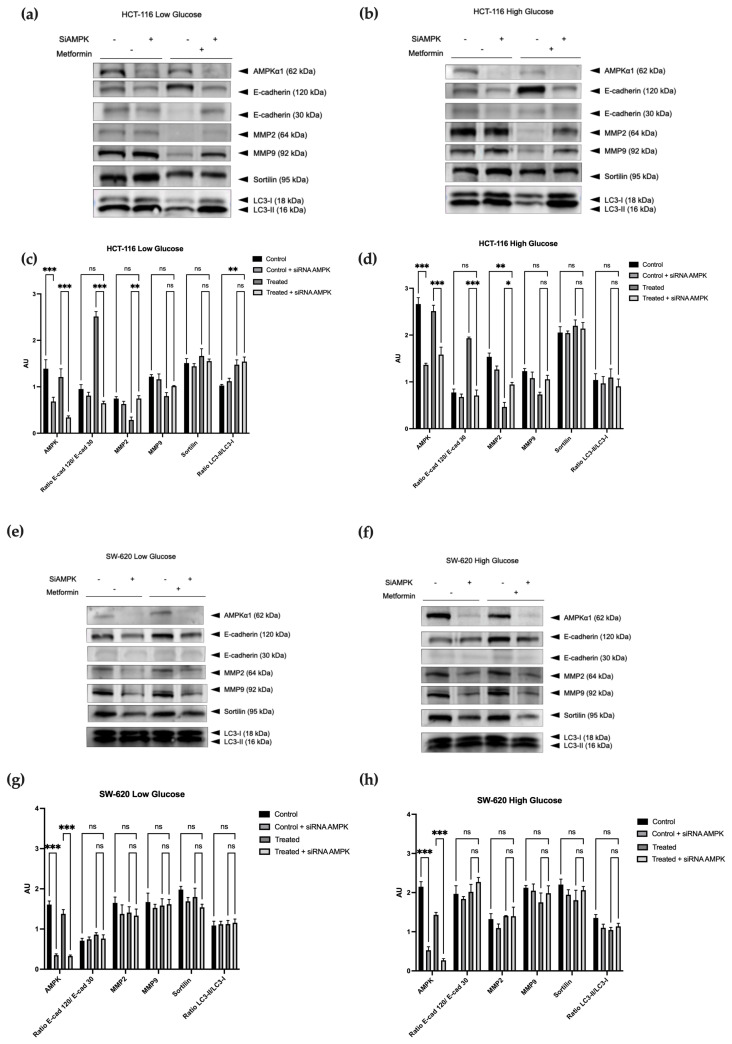
Effect of AMPKα1 silencing on metformin treatment. The expression of AMPK was silenced by siRNA in HCT-116 (**a**–**d**) and SW-620 (**e**–**h**). Cells were treated (or not) with metformin and the expression of indicated proteins was evaluated by Western blot. The TotalStain Q (Appendix A) allowed us to evaluate the equal loading of protein in order to achieve quantification. Performed in biological triplicate. * *p* < 0.05; ** *p* < 0.01; *** *p* < 0.001. ns: no significance. The uncropped blots are shown in Appendix A.

**Figure 5 cancers-15-03724-f005:**
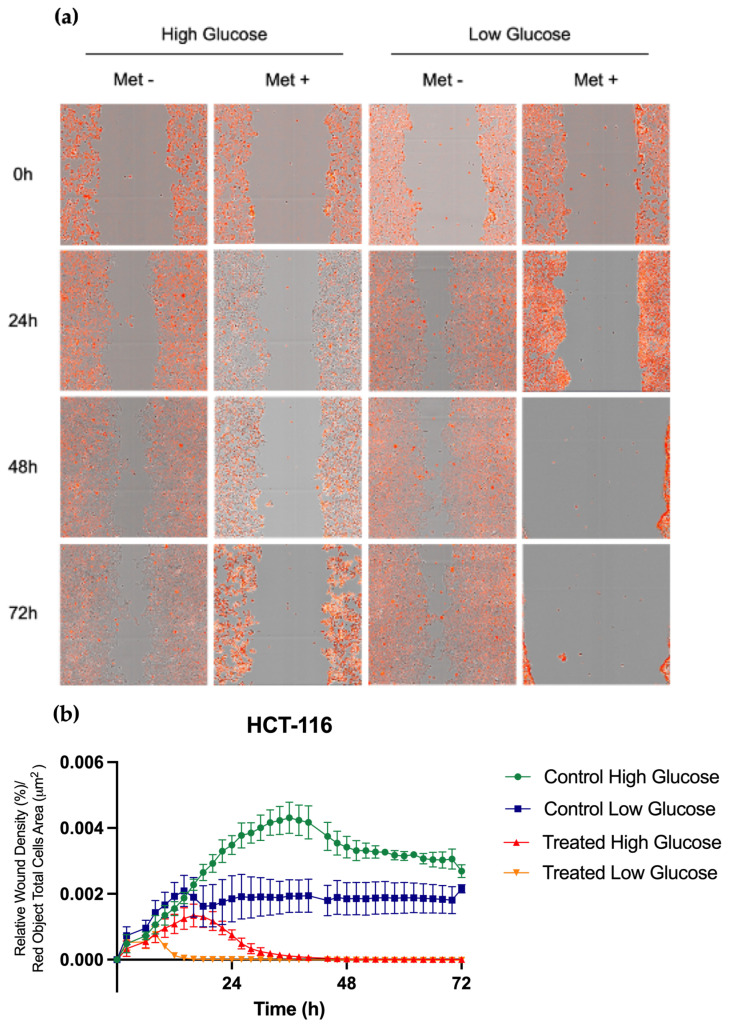
Impact of metformin on HCT-116 migration, depending on glucose concentrations. (**a**) HCT-116 cells were treated with metformin for 72 h. (**b**) Relative wound density was quantified by the wound-healing assay using Incucyte software and normalized by the red object’s total cell area. Performed in biological triplicate.

**Figure 6 cancers-15-03724-f006:**
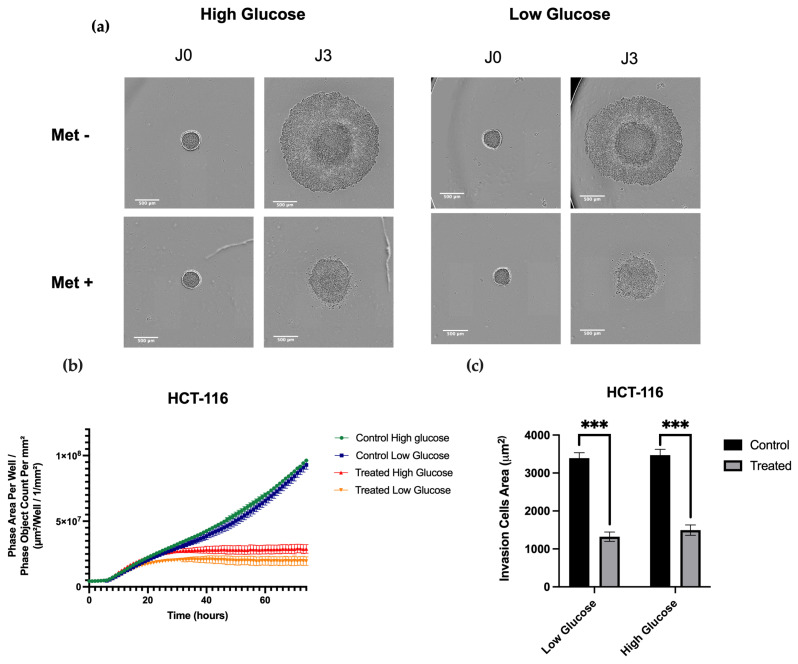
Impact of metformin on HCT-116 invasion, depending on glucose concentrations. (**a**) HCT-116 spheroids were treated or not with metformin for 72 h. (**b**) The area of the spheroids was quantified by Incucyte software and performed in biological triplicate. (**c**) The invasion of the spheroids was quantified by ImageJ (1.53a). Performed in biological triplicate. *** *p* < 0.001.

**Figure 7 cancers-15-03724-f007:**
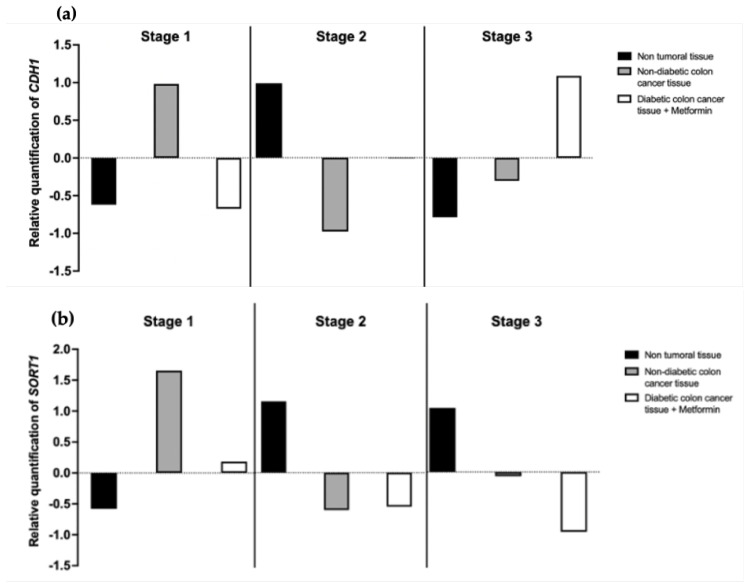
Metformin’s effect on CDH1 and SORT1 in a cohort of CC patients. (**a**) The tissues were lysed and the CDH1 gene (**a**) or the SORT1 gene (**b**) were quantified by real-time qPCR, after treatment with IC 50 metformin for 48 h in two glucose concentrations, in reference to ACTB for normalization. Each column shows the z-score, performed in technical triplicate.

**Figure 8 cancers-15-03724-f008:**
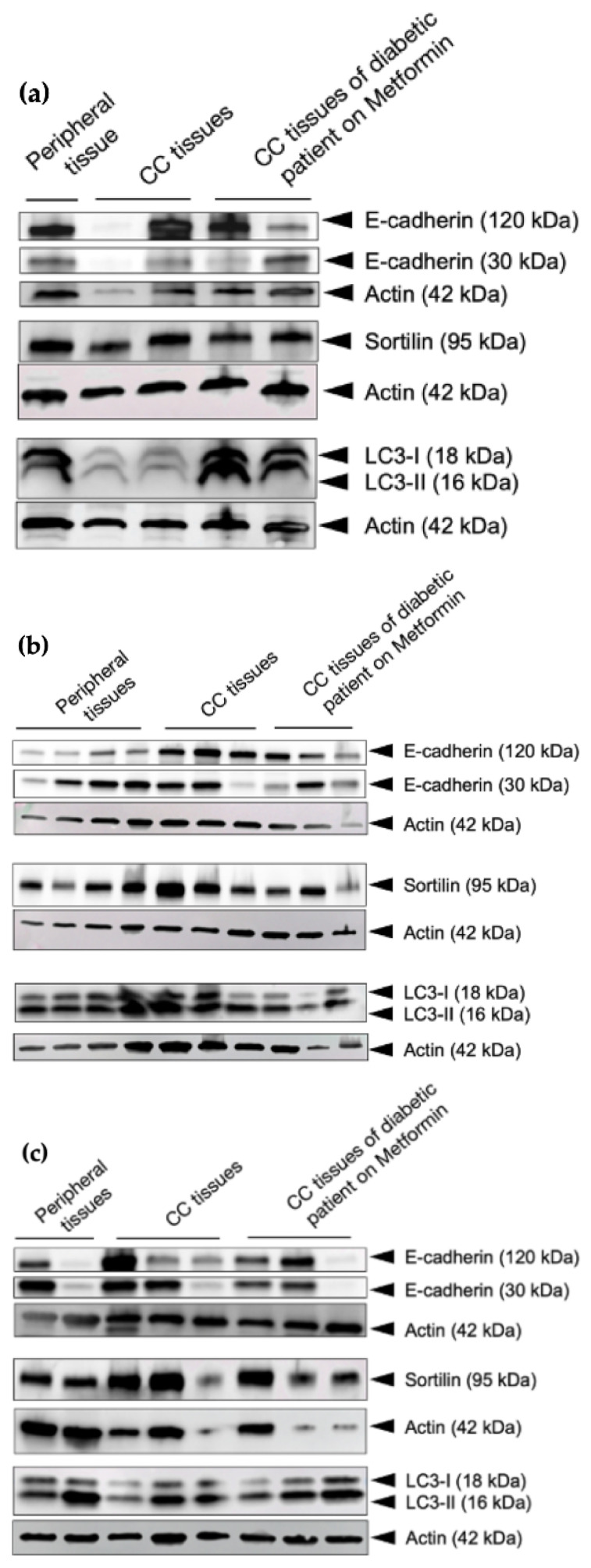
Metformin’s effect on E-cadherin, sortilin, and LC3-II expressions in a cohort of CC patients. Western blot of E-cadherin, sortilin, and LC3 on (**a**) **Stage 1** patients (**b**) **Stage 2** patients (**c**) **Stage 3** patients. Actin, which is considered a control, confirms the integrity and equal loading of protein. The uncropped blots are shown in Appendix A.

**Figure 9 cancers-15-03724-f009:**
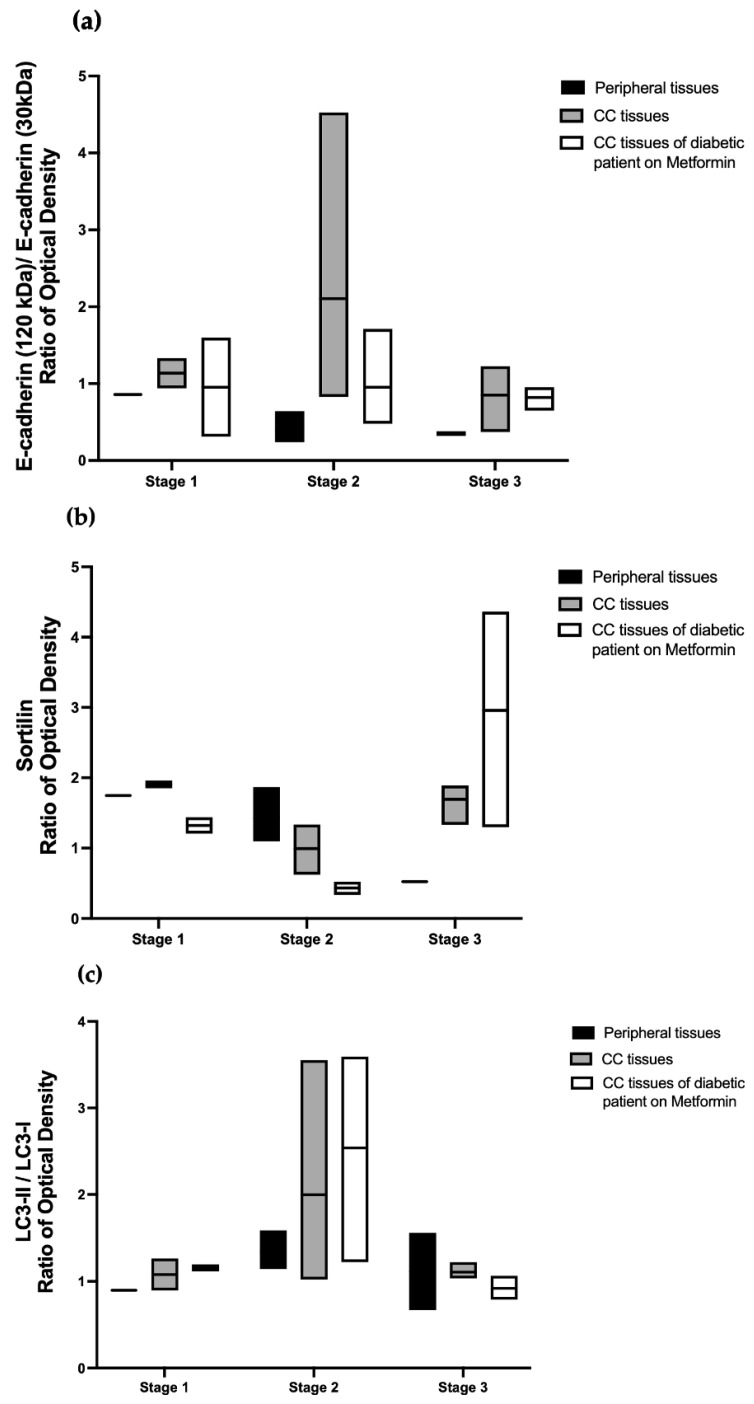
Metformin’s effect on E-cadherin, sortilin, and LC3-II expressions in a cohort of CC patients. Box representation (medium and line) of (**a**) The ratio of E-cadherin (120 kDa) and the total form of E-cadherin on E-cadherin 30 kDa, the aggressive form; (**b**) sortilin; (**c**) LC3-II/ LC3-I ratio.

**Table 1 cancers-15-03724-t001:** Main characteristics of the 2 colon cancer (CC) cell lines used.

Colorectal Cancer Cell Line	HCT-116	SW-620
Type	Colon adenocarcinoma	Lymph node metastasis from a colon adenocarcinoma
Stage	Early stage	Advanced stage
CMS	CMS1	CMS4
TP53	wt	p.R273H; p.P309S
KRAS	p.G13D	p.G12V
BRAF	wt	wt
PIK3CA	p.H1047R	wt
PTEN	wt	wt
MSI	MSI	MSS
CIMP	CIMP+	CIMP−

**Table 2 cancers-15-03724-t002:** Patients’ characteristics of the cohort.

Colon Cancer (CC)		Peritumoral Tissues	Non Diabetic CC Samples (without Metformin)	Diabetic CCs + Metformin
Cohort size	n	7	8	8
Sex	males	3	2	5
	females	4	6	3
Age	means (+/−SEM)	72 (+/−15.8)	75 (+/−7.9)	72 (+/−7.1)
	median (min-max)	79 (48–88)	75 (60–85)	72 (65–84)
Age Class	≤72 years	2	3	4
	≥72 years	5	5	4
Stage	1	1	2	2
	2	4	4	3
	3	2	3	2

**Table 3 cancers-15-03724-t003:** References of the primers used.

Name	Gene Name	References
*CDH1*	Cadherin-1	Hs01023894_m1
*SORT1*	Sortilin-1	Hs00234110_CE
*ACTB*	Actin beta	Hs00368836_CE

**Table 4 cancers-15-03724-t004:** References of the primary antibody used.

Primary Antibody	References	Antibody Isotype	Dilution
AMPKα1	Cell Signaling, Danvers, MA, USA (2795)	Rabbit, San Francisco, CA, USA	1:1000
E-cadherin	BD Biosciences, Franklin Lakes, NJ, USA (610182)	Mouse	1:1000
MMP2	Cell Signaling (4022)	Rabbit	1:1000
MMP9	Abcam, Boston, MA, USA (ab38898)	Rabbit	1:1000
Sortilin	Abcam (ab16640)	Rabbit	1:1000
LC3-B	Cell Signaling (2775)	Rabbit	1:1000
Actin	Sigma Aldrich, St. Louis, MO, USA (A5441)	Mouse	1:10,000

## Data Availability

The data presented in this study are available in this article (and Appendix A).

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
