# Peer review of "Metformin Treatment Reduces CRC Aggressiveness in a Glucose-Independent Manner: An In Vitro and Ex Vivo Study"

_cancers, 2023, doi:10.3390/cancers15143724_

Round 1

Reviewer 1 Report

The ms by Boutaud et al. presents the results of a study on influence of metformin on CRC aggressiveness. The study is well designed and conducted.

I would suggest some corrections:

1. The patients' group is the weakest point of the research. The group is very small, moreover it is divided into 3 subgroups with less than 10. As a part of the whole range of experiments it can stay, but is not sufficient to make any definite conclusions (esp. like those in lines 305-310). Therefore it needs to be better discussed and referenced.

2. SD and significance is not present on Figure 2b) charts. It is also not present on Fig. 4 and 6 - please correct/explain.

3. Some small punctuation/spelling/grammatical/styllistic mistakes (like "..." line 299, "in our previous experience" 299-300 ? - rather our previous research/data; and few others) - need one thorough reading.

Author Response

The ms by Boutaud et al. presents the results of a study on influence of metformin on CRC aggressiveness. The study is well designed and conducted.

I would suggest some corrections:

  1. The patients' group is the weakest point of the research. The group is very small, moreover it is divided into 3 subgroups with less than 10. As a part of the whole range of experiments it can stay, but is not sufficient to make any definite conclusions (esp. like those in lines 305-310). Therefore it needs to be better discussed and referenced.

We completely agree with the reviewer and we are awaked of the weakness of this part in our study, which justifies our use of cautious terms. As recommended, we further minimized conclusion drawn from patient’s samples analysis. Furthermore, we emphasized the discussion with additional references.

  1. SD and significance is not present on Figure 2b) charts. It is also not present on Fig. 4 and 6 - please correct/explain.

We introduced statistical analysis with mark in the new figure 3 (ex figure 2). For this figure, the loading control was done using the totalstain Q from azure biosystems, instead of actin or tubulin. All controls are shown in figure S2, to keep the main manuscript light. Furthermore, as requested by another referee (#3), we added western blot concerning metalloprotease MMP2 and MMP9 (involved in cadherin cleavage). Of course, statistical analysis is also reported in the new figure.

For Fig 4 and Fig 6, which become in the revised manuscript Fig 7 and 9, established with data from patients (and as underlined by the reviewer in comment 1), the weak number of cases in each groups did not allow us to reach any reliable conclusions. So we preferred to exhibit raw results, without statistical analysis.

  1. Some small punctuation/spelling/grammatical/styllistic mistakes (like "..." line 299, "in our previous experience" 299-300 ? - rather our previous research/data; and few others) - need one thorough reading.

We apologize for persistent mistakes that we have corrected in the new submitted version.

Reviewer 2 Report

In this manuscript authors evaluated the metformin-mediated effect on colorectal cancer aggressiveness according to different glucose levels conditions.

Although the manuscript is interesting and generally well written, it presents some important flaws. In particular:

Lines 42-50: it deserves to be pointed out that  the anti-tumor effect of metformin is also due to its capacity to restore chemotherapeutics sensitivity in cancer cells (see PMID: 36361682 )

2.6. Western Blotting: it would be better listing the primary antibodies used in a dedicate table. Moreover, primary antibodies dilutions must be reported.

Figures: significant differences between groups must be shown with asterisks or similar

Figure 2: western blot showing ACTB must be reported

Figure 3: cell confluence is not appropriated to perform wound healing essay since it is very low and cells can move everywere not only in the wound 

An accurate revision of typing errors is recommended

Author Response

In this manuscript authors evaluated the metformin-mediated effect on colorectal cancer aggressiveness according to different glucose levels conditions.

Although the manuscript is interesting and generally well written, it presents some important flaws. In particular:

Lines 42-50: it deserves to be pointed out that  the anti-tumor effect of metformin is also due to its capacity to restore chemotherapeutics sensitivity in cancer cells (see PMID: 36361682 )

Indeed, we agree with the reviewer, since spectrum of metformin's effects on cancer cells is very wide-ranging. We added  more references to argue the point on synergistic effect between metformin and chemotherapies.

2.6. Western Blotting: it would be better listing the primary antibodies used in a dedicate table. Moreover, primary antibodies dilutions must be reported.

As requested, we replaced the text by a dedicated table (table 4), where primary antibodies and working dilutions are reported.

Figures: significant differences between groups must be shown with asterisks or similar

We introduced statistical analysis with mark in the new figure 3 (ex figure 2). For this figure, the loading control was done using the totalstain Q from azure biosystems, instead of actin or tubulin. All controls are shown in figure S2, to keep the main manuscript light. Furthermore, as requested by another referee (#3), we added western blot concerning metalloprotease MMP2 and MMP9 (involved in cadherin cleavage). Of course, statistical analysis is also reported in the new figure

For Fig 4 and Fig 6, which become in the revised manuscript Fig 7 and 9, established with data from patients (and as underlined by the reviewer in comment 1), the weak number of cases in each groups did not allow us to reach any reliable conclusions. So we preferred to exhibit raw results, without statistical analysis.

Figure 2: western blot showing ACTB must be reported

In this figure (now fig 3) and in new figure 4, the western blot normalization was achieved using quantitative fluorescent total protein stain (totalstain Q from azure biosystems). The pictures of total staining were shown respectively in figure S2 and S3. Additional explanation was added in material and methods section.

Figure 3: cell confluence is not appropriated to perform wound healing essay since it is very low and cells can move everywere not only in the wound 

Since the reviewer seemed to be not convinced by the wound healing assay, and according to a request from another reviewer (#3), we performed a Matrigel cell invasion assay that we added in figure 6. Results we obtained corroborated data from migration assessed by wound healing assay, showing that metformin reduces the invasive property of CRC cells (in vitro). We hope this experiment will provide an appropriate complement to our observations.

An accurate revision of typing errors is recommended

We apologize for persistent mistakes that we have corrected in the new submitted version.

Reviewer 3 Report

 This study discusses the results of an in vitro investigation into the effects of Metformin on HCT-116 and SW-620 colon cancer cell lines. The authors examined the effect of low and high glucose concentrations on cellular growth and the efficacy of Metformin in inhibiting cell growth. Metformin in dose-dependently inhibited cell proliferation in both cell lines, and the half-maximal inhibitory concentration (IC50) was determined. The researchers also examined the effect of Metformin on the expression of aggressiveness markers E-cadherin, Sortilin, and LC3-II. Metformin treatment decreased the aggressiveness of early-stage colon cancer cells, but had no effect on advanced-stage cells regardless of glucose concentration. Regardless of glucose concentration, the study found that Metformin inhibited the migration of early-stage colon cancer cells.

Here are some recommendations for experiments that could enhance the quality of the manuscript:

1-Metformin has been reported to induce apoptosis in some cancer cell types; therefore, it would be beneficial to examine whether this mechanism also affects to the HCT-116 and SW-620 cell lines used in this investigation. This could be accomplished with techniques such as Annexin V/PI staining and flow cytometry analysis.

2-Investigate the role of AMPK in Metformin-mediated effects. Although the authors described that Metformin activates the AMPK pathway, they do not investigate its specific role in the observed alterations in gene and protein expression. Additional investigations, such as siRNA-mediated AMPK knockdown, could help determine whether the observed effects are mediated directly or indirectly by this pathway.

3-The study concentrates on the effects of Metformin on EMT, autophagy, and Sortilin expression, but there are additional aggressiveness markers that could be investigated. For instance, the authors could examine the impact of Metformin on the expression of matrix metalloproteinases (MMPs), which are implicated in tumor invasion and metastasis.

4-The authors treated the cells with Metformin for 48 hours before analyzing the alterations in gene and protein expression. However, it would be beneficial to determine whether the effects of extended or shorter treatment durations vary. This could aid in determining the optimal treatment regimen for future clinical applications.

5- Cell Invasion assay also should be added in the figure 2.

Author Response

Here are some recommendations for experiments that could enhance the quality of the manuscript:

1-Metformin has been reported to induce apoptosis in some cancer cell types; therefore, it would be beneficial to examine whether this mechanism also affects to the HCT-116 and SW-620 cell lines used in this investigation. This could be accomplished with techniques such as Annexin V/PI staining and flow cytometry analysis.

We agree with the reviewer when saying that metformin could induce apoptotic cell death. So we investigate this feature using Incucyte, where apoptotic cells were labelled with Incucyte® Annexin V Dyes for apoptosis and proliferation was evaluated by cell-by-cell analysis software module. These results appear now in figure 2.

2-Investigate the role of AMPK in Metformin-mediated effects. Although the authors described that Metformin activates the AMPK pathway, they do not investigate its specific role in the observed alterations in gene and protein expression. Additional investigations, such as siRNA-mediated AMPK knockdown, could help determine whether the observed effects are mediated directly or indirectly by this pathway.

We thank the reviewer for this comment and for the opportunity to improve our manuscript. Indeed, AMPK is a well-known target of metformin and it is important to investigate its role in the cell response to such a treatment. We performed inhibition of AMPK expression by si-RNA and we evaluated expression of proteins of interest i.e. E-cadherin, Sortilin, LC3, MMP2 and MMP9. This is now introduced as figure 4 and loading controls, realized with totalstain Q from azure biosystems, are available in figure S3. Results are discussed in the dedicated section.

3-The study concentrates on the effects of Metformin on EMT, autophagy, and Sortilin expression, but there are additional aggressiveness markers that could be investigated. For instance, the authors could examine the impact of Metformin on the expression of matrix metalloproteinases (MMPs), which are implicated in tumor invasion and metastasis.

We agree with reviewer when saying that aggressiveness of cancer cells is a complex process, implicating various mechanisms. Among them, and in relation with our focused interest on E-cadherin, we performed and added WB for MM2 and MMP9 in figure 3, since they are involved in E-Cad cleavage (ref 32 ; 10.2741/4031 ; 10.18632/oncotarget.1463 )

4-The authors treated the cells with Metformin for 48 hours before analyzing the alterations in gene and protein expression. However, it would be beneficial to determine whether the effects of extended or shorter treatment durations vary. This could aid in determining the optimal treatment regimen for future clinical applications.

As requested by the reviewer, we extended time treatment with metformin by testing cytotoxicity of the molecule at 24 and 72h. This analysis is now included in figure 1. In the further work, the cells were treated at the IC50 determined at time 48h.

5- Cell Invasion assay also should be added in the figure 2.

According to the request of two reviewers (2 and 3), we performed a Matrigel cell invasion assay that we added in figure 6. Results we obtained corroborated data from migration assessed by wound healing assay, showing that metformin reduces the invasive property of CRC cells (in vitro). We hope this experiment will provide an appropriate complement to our observations.

Round 2

Reviewer 2 Report

the manuscript has been significantly improved and can be accepted in the present form 

Reviewer 3 Report

The Author have done all experiments and corrections suggested by reviewers.